# Evaluation of albumin use in a community hospital setting: A retrospective study looking at appropriate use and prescribing patterns

Timothy Coyle[1], Samuel M. John[2]*

1 PGY-2 Medication Use Safety and Policy, Wellstar Health System, Marietta, GA, United States of America, 2 Department of Pharmacy Practice, PGY-1 Residency Program, PCOM Georgia, Suwanee, GA, United States of America

* SamuelJo@pcom.edu

**Data Availability Statement:** All relevant data are within the paper and its supporting information files.

**Funding:** The authors received no specific funding for this work.

## Abstract

### Purpose

Albumin has been shown to be safe and effective in clinical practice for a wide variety of indications. The purpose of this medication use evaluation is to quantify the use of albumin in the community hospital setting based on indication and prescribing department.

### Methods

This study is a retrospective, single-center, chart review over a 6-month period of 186 patients aged 18 and older who were treated with IV human albumin 5% or 25% at a single 202-bed community hospital setting from February 1, 2020, to August 1, 2020. A chart review was completed for each patient and the data collected included date of albumin administration, the ordering provider, the specialty of the provider, the indication for albumin as stated in the order, patient notes, crystalloid therapy use prior to albumin, albumin strength, the presence of acute or chronic renal, hepatic or respiratory disorders, and lab values denoting renal and hepatic function. Appropriate albumin use was determined utilizing criteria which included FDA labeled indications, the Surviving Sepsis Campaign, and existing literature.

### Results

A total of 186 patients received albumin 5% or 25% IV solution at least once during the study period. The study population was 52.2% female, and the average age was 68 years. Of the patients selected for the study, 23 (11.6%) had chronic hepatic disease, and 37 (18.7%) had chronic renal disease. The top indications for which albumin was administered were sepsis or septic shock (25.3%), hypotension or hypovolemia (19.4%), intra-dialytic hypotension (13.4%), fluid support in surgery (10.8%), and nephrosis or nephropathy (10.8%). The departments with highest albumin use during this study period were critical care (41%), nephrology (28%), and surgery (17%). Overall, albumin was used for an appropriate indication in 126 out of 186 patients (67.7%).

**Competing interests:** The authors have declared that no competing interests exist.

## Conclusion

We found that albumin was most utilized for sepsis and septic shock, hypovolemia and hypotension, and intradialytic hypotension in our community hospital setting and it was most frequently ordered by critical care, nephrology, and surgical departments. Further research could determine if this trend is seen in other community hospital settings.

## Introduction

Albumin has been used clinically for multiple indications, including fluid resuscitation and support in pulmonary, renal and hepatic conditions [1,2]. It is associated with higher cost of therapy compared to crystalloids, making its use somewhat controversial there is no comparable benefit of using albumin over crystalloids [3]. The Saline versus Albumin Fluid Evaluation (SAFE) trail in 2004 showed that 4% albumin was comparable to normal saline when used for resuscitation [2,3]. In recent years, more guidelines for albumin use have been utilized in multiple healthcare systems, even though clinical benefit of using albumin is unclear [4,5].

The current FDA-labeled indications for albumin are acute respiratory distress syndrome, cirrhotic ascites, erythrocyte resuspension, hypovolemia, neonatal hemolytic disease, and adjunct treatment for nephrosis in combination with diuretics.[1] Albumin is also included in the Surviving Sepsis campaign for patients that are hypovolemic that require large volumes of crystalloids [6,7]. There are multiple off-label uses of albumin within an acute care setting, and variability of albumin use remains high between providers, regardless of the presence of guidelines for appropriate use [8].

Multiple studies have shown the cost-reduction potential of restricting albumin use, but these were focused on the critical care setting, and there is less data for the overall albumin use within a community hospital setting [7,9]. The purpose of this medication use evaluation is to quantify the use of albumin in a community hospital setting. The results could be used to describe the use of albumin by department and to provide information regarding the departments that utilize albumin the most in the current healthcare setting.

## Objectives

The primary objective was to determine the prescribing practices of albumin use by indication for a community hospital setting. The secondary objective was to determine appropriate use of albumin based on existing literature, including the FDA labeled indications, indications from the Surviving Sepsis Campaign and from the study conducted by Buckley et al regarding pharmacist intervention in albumin use [1,6,9].

## Materials and methods

This is a retrospective, single-center, chart review over a 6-month period from February 1, 2020, to August 1, 2020 of 186 patients who were treated with IV human albumin 5% or 25% at a community hospital (Wellstar North Fulton Hospital) consisting of 202 patient beds, 27 of which are adult ICU beds. At the time of this study, the hospital services included a level-2 trauma center, adult ICU, and medical and surgical floors. This study was evaluated by the Wellstar Research Institute and determined to be waived from the institutional review board. After collection of data from the patients' charts, the patients' information was anonymized prior to evaluation of the data. Patients were included if they were aged 18 or older and

received at least one dose of albumin during their time in the hospital. Patients were excluded if they did not receive a dose of albumin during their hospital stay. Descriptive analysis including frequencies and percentages of all requested variables was performed, using mean and standard deviation, as appropriate. This was a descriptive chart review, and no statistical tests were employed.

A medication report was generated in the electronic medical record software, Epic®, for patients within the hospital who received IV albumin during their admission between the dates February 1, 2020, to August 1, 2020. A chart review of the patients was then conducted in which the indication for albumin use was determined from the order that was entered, from the note from the provider or a note on which the ordering provider was the co-signer, and through the medication profile review. The data collected included patient demographics including age, gender ethnicity, and chronic hepatic or renal disease, if any. Additional data included the date albumin was first administered, the ordering provider, the specialty of the ordering provider, the indication or interpreted indication of albumin, crystalloid therapy prior to the use of albumin, albumin strength, the presence of acute or chronic renal, hepatic, or respiratory disease, as included in the previous medical history of the patient chart, and renal and hepatic lab values (SCr, BUN, eGFR, AST, ALT, serum albumin, total bilirubin).

Appropriate indications of albumin were determined utilizing a criteria that was made using the FDA-approved indications, the Surviving Sepsis Campaign, and the indications considered appropriate by Buckley, et al [1,6,9]. The uses of albumin that were considered appropriate included acute respiratory distress syndrome, cirrhotic ascites, hypovolemia, nephrosis or nephropathy, fluid resuscitation in septic shock in combination with crystalloid therapy, cardiovascular surgery, large volume paracentesis greater than five liters, hepatorenal syndrome, and therapeutic plasma exchange [1,6,9]. For indications that were outside of these criteria, the use was considered inappropriate.

## Results

A total of 194 patients were screened. Of these patients, 186 received albumin 5% or 25% IV solution at least once during the study period. Table 1 describes baseline characteristics of the patient population. The study population was 52.2% female and had an average age of 68 years. One-hundred and twenty-two patients (65.6%) identified as white or Caucasian, 30 patients (16.1%)

**Table 1. Baseline characteristics.**

| Baseline Characteristics (n = 186) | |
|---|---|
| Male, n (%) | 89 (47.8%) |
| Female, n (%) | 97 (52.2%) |
| Average Age, years ± SD | 68 ± 15 |
| Ethnicity, n (%) | |
| White/Caucasian | 122 (65.6%) |
| African American | 30 (16.1%) |
| Hispanic or Latino | 15 (8.1%) |
| Asian | 5 (2.7%) |
| Other | 14 (7.5%) |
| Chronic Hepatic Disease, n (%) | 23 (11.6%) |
| Chronic Renal Disease, n (%) | 37 (18.7%) |
| Albumin Product Used, n (%) | |
| 5% | 46 (24.7%) |
| 25% | 140 (75.2%) |

identified as African American, and 15 patients (8.1%) identified as Hispanic or Latino. Of the patients selected for the study, 23 (11.6%) had chronic hepatic disease, and 37 (18.7%) had underlying chronic renal disease. Most of the patients (75.2%) received albumin 25% solution.

The indications for albumin use with the largest number of patients over the 6-month study period were sepsis or septic shock (25.3%), hypotension or hypovolemia (19.4%), intra-dialytic hypotension (13.4%), fluid support in surgery (10.8%), and nephrosis or nephropathy (10.8%) (Table 2). Other indications representing less than 5% of the albumin use included acute respiratory distress syndrome (ARDS), paracentesis, hyponatremia, hypervolemia or edema, cirrhosis, hepatorenal syndrome, respiratory failure, and hypoalbuminemia in liver disease.

Overall, 126 out of 186 patients (67.7%) in this study were administered albumin for an appropriate indication as defined by our criteria utilizing the FDA-labeled indications, Surviving Sepsis Campaign guidelines, and the study by Buckley and colleagues [9]. Albumin was used appropriately in the 47 patients presenting with sepsis or septic shock, the 36 patients presenting with hypotension or hypovolemia, and the 20 patients presenting with nephrosis or nephropathy. Albumin was considered inappropriate in the setting of dialysis and fluid support in surgery. Four of the 7 patients with paracentesis were administered albumin appropriately, and three were not due to volume of paracentesis being below 5 liters. Other appropriate indications as defined by the criteria included cirrhosis, hepatorenal syndrome, hypoalbuminemia in liver disease, hepatic hydrothorax, plasmapheresis, and therapeutic plasma exchange. Other inappropriate indications for which albumin was used were hyponatremia, hypervolemia, and respiratory failure (Fig 1).

Most of the orders for albumin were prescribed by providers specializing in critical care, nephrology, or surgery. Fig 2 shows the prescribing patterns of physicians in the community hospital setting based on specialty of the prescriber. Of the patients that received albumin in this study, 41% were given albumin as ordered by critical care, 28% of patients were administered albumin from nephrology orders, and 17% were administered albumin from surgical orders. Other departments that prescribed and administered albumin in this study period

**Table 2. Albumin use by indication.**

| Indication | Number of Patients Who Received One or More Doses of Albumin, n (%) |
|---|---|
| Sepsis/Septic Shock | 47 (25.3%) |
| Hypovolemia/Hypotension | 36 (19.4%) |
| Intra-dialytic Hypotension | 25 (13.4%) |
| Fluid Support in Surgery | 20 (10.8%) |
| Nephrosis/Nephropathy | 20 (10.8%) |
| Paracentesis | 7 (3.8%) |
| ARDS | 7 (3.8%) |
| Hyponatremia | 6 (3.2%) |
| Hypervolemia | 4 (2.2%) |
| Cirrhosis | 3 (1.6%) |
| Hepatorenal Syndrome | 3 (1.6%) |
| Respiratory Failure | 2 (1.1%) |
| Hypoalbuminemia in Liver Disease | 2 (1.1%) |
| Hepatic Hydrothorax | 1 (0.5%) |
| Plasmapheresis | 1 (0.5%) |
| Thoracentesis | 1 (0.5%) |
| Therapeutic Plasma Exchange | 1 (0.5%) |
| Total | 186 |

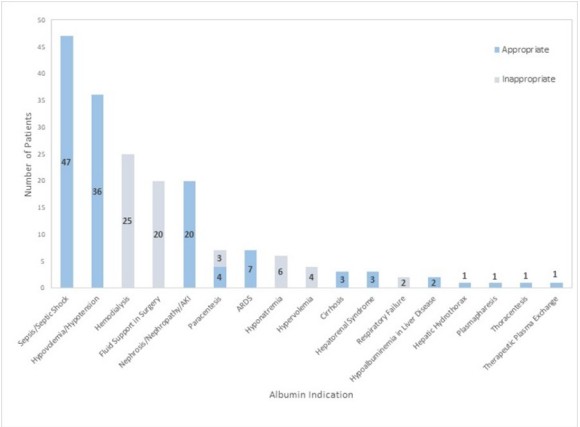

**Fig 1. Albumin use by initial indication (February 1, 2020–August 1, 2020).** AKI indicates acute kidney injury, ARDS indicates acute respiratory distress syndrome.

include hospital medicine, gastroenterology, cardiology, respiratory medicine, emergency medicine, and gynecology/oncology.

## Discussion

Our study showed that albumin was used for an appropriate indication in 126 out of 186 patients (67.7%). The criteria for appropriate use were created by utilizing FDA approved indications, the Surviving Sepsis Campaign and the previously completed study completed by Buckley et al [1,6,9]. The criteria for appropriate use were intentionally made to be broad to account for provider experience and variability. We used the study completed by Buckley, et al, to build our criteria due to the pharmacist-led strategy that their method employed [9]. This would also help to see if criteria for appropriate use of albumin is applicable between practice sites [9].

Albumin is considered appropriate when used in the context of sepsis and septic shock after crystalloid therapy is given first. The evidence for albumin benefit in sepsis or septic shock is unclear [10,11]. The study completed by Caironi et al and showed that there was no difference in mortality for patients with septic shock between patients receiving albumin versus normal saline, but a meta-analysis by Xu et al showed a trend toward lower 90-day mortality in severe sepsis patients treated with albumin versus crystalloid therapy [10,11]. This suggests further studies still need to be performed to establish any significant benefit of albumin over crystalloid therapy [10,11].

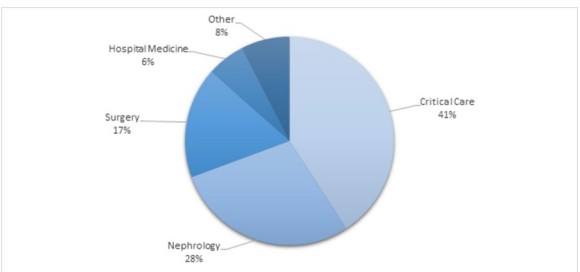

**Fig 2. Albumin administrations as prescribed by specialty.** Other: Gastroenterology, physical medicine, cardiology, respiratory, emergency medicine, gynecology oncology.

In our study, we saw more albumin use in the context of nephrology. Second to critical care, it was the highest practice that wrote orders for albumin. While it can be used in patients with hepatorenal syndrome to increase renal function, it can also be used in patients with nephrosis or nephrotic syndrome may warrant future studies [12]. In one study, furosemide with albumin produced favorable results, increasing the pharmacologic effects of furosemide, though further studies should be performed [13].

The area in which albumin use was considered inappropriate in this study was fluid support in the setting of non-cardiovascular surgery. The surgical procedures that utilized albumin included genitourinary, hip, ovarian, neurologic, or trauma surgeries. According to the current literature, the preferred solutions for fluid resuscitation in these types of surgeries are crystalloids [9,10]. It should be noted, however, that the hospital procedure for this site permit albumin to be used in the surgical setting if crystalloid therapy is utilized first. Further studies could analyze albumin use in the surgical setting.

Albumin use was also considered inappropriate in the setting of intradialytic hypotension [1,9]. Although off-label use exists for fluid support during dialysis, the criteria utilized in this study did not include dialysis as an appropriate indication. The hospital order set, however, allows for albumin to be utilized for patients receiving hemodialysis if they were hypotensive despite crystalloid therapy alone, which may be a reason that use was high in the setting of hemodialysis.

The albumin use in this study is comparable to other studies of albumin use within a hospital setting. The study by Castillo, et al found that approximately 45% of albumin was prescribed inappropriately according to their guidelines [14]. Similarly, the study conducted by Buckley et al showed that up to 63.4% of albumin administered was considered inappropriate, although their criteria was more stringent [9].

There is potential for pharmacist intervention in reducing inappropriate albumin use. Tigabu, et al, performed a cost analysis of albumin in septic shock that found that albumin use increased the cost of medication therapy without improving the 28-day mortality compared to normal saline [15]. Additionally, Buckley, et al, found that their pharmacist-led strategy led to a 50.9% decrease of inappropriate use of albumin [9]. Future studies could include the goal of developing criteria for appropriate albumin use within the hospital system.

One of the limitations of this study was the retrospective nature and reliance on chart review as the primary data collection method. Another limitation is the wide variety of indications albumin, leading to some ambiguity in determining the indication for albumin, though every effort was made to review the clinical notes, orders, and medication profile to determine the most likely indication.

## Conclusion

We found that albumin was most utilized for sepsis and septic shock, hypovolemia and hypotension, and intradialytic hypotension in our community hospital setting and it was ordered by critical care, nephrology, and surgery departments. This study represents a single community hospital site, and further research could be conducted to determine if these trends are seen in other community hospital settings.

## Supporting information

**S1 Dataset.**
(XLSX)

## Author Contributions

**Conceptualization:** Samuel M. John.

**Data curation:** Timothy Coyle.

**Formal analysis:** Timothy Coyle.

**Investigation:** Timothy Coyle.

**Methodology:** Timothy Coyle.

**Project administration:** Timothy Coyle.

**Resources:** Samuel M. John.

**Writing – original draft:** Timothy Coyle.

**Writing – review & editing:** Samuel M. John.

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
