## [Decision Letter · Decision Letter 0]

21 Jul 2021

PONE-D-21-19973

Evaluation of Albumin use in a Community Hospital Setting: A retrospective study looking at appropriate use and prescribing patterns 

PLOS ONE

Dear Dr. JOHN,

Thank you for submitting your manuscript to PLOS ONE. After careful consideration, we feel that it has merit but does not fully meet PLOS ONE’s publication criteria as it currently stands. Therefore, we invite you to submit a revised version of the manuscript that addresses the points raised during the review process.

We look forward to receiving your revised manuscript.

Kind regards,

Aleksandar R. Zivkovic

Academic Editor

PLOS ONE

2. In ethics statement in the manuscript and in the online submission form, please provide additional information about the patient records used in your retrospective study. Specifically, please ensure that you have discussed whether all data were fully anonymized before you accessed them and/or whether the IRB or ethics committee waived the requirement for informed consent. If patients provided informed written consent to have data from their medical records used in research, please include this information.

3. "In your Data Availability statement, you have not specified where the minimal data set underlying the results described in your manuscript can be found. PLOS defines a study's minimal data set as the underlying data used to reach the conclusions drawn in the manuscript and any additional data required to replicate the reported study findings in their entirety. All PLOS journals require that the minimal data set be made fully available. For more information about our data policy, please see http://journals.plos.org/plosone/s/data-availability.

6. We note you have included a table to which you do not refer in the text of your manuscript. Please ensure that you refer to Table 1 in your text; if accepted, production will need this reference to link the reader to the Table.

Reviewers' comments:

Reviewer #1: Thank you for asking me to review this manuscript.

Samuel and Coyle described the use of human albumin solution (HAS) in a community hospital.

Overall, the manuscript is a clearly written however I am uncertain as to why the results warrant a publication. In its present form, the work resembles an audit rather than original research.

The introduction does not paint a convincing picture as to why the research question is important. Furthermore, why should prescribing practices of HAS vary between tertiary hospitals and community hospitals if FDA approval for HAS is standard; the authors do not discuss how their results differ from that of current literature. As such, I struggle to see where the novelty lies.

To be considered for publication, the manuscript is in need of a Major revision. The introduction and discussion needs to be improved and more evidence provided. 9 references is insufficient for an original research article.

My other comments are listed below and I hope the authors find them useful. I wish the authors all the best in their revisions and future endeavours.

Introduction

"The results could be used to influence prescribing practices in the health care system and decrease cost to the hospital and the patient."

I am unsure how describing the use of HAS in a community hospital can influence "prescribing practices in the health care system and decrease cost to the hospital and patient". It is not clearly explained how this can be achieved with the results herein.

Methods

There is mention of standard deviations although I do not reported in the results.

Observational studies should be reported using the STROBE checklist. There are missing segments in the methods e.g. inclusion and exclusion criteria.

Why wasn't data on the adverse reactions of HAS investigate? Were there any differences in the approved and "off label" use?

Results

I do not see how these results are different from data that is already available in literature.

Discussion

The discussion is very superficial and not impactful.

Reviewer #2: In a single-center community hospital study, therapeutic use of human albumin solution was analysed. Appropriate use was defined according to labelled indications and SCC Guideline recommendation. Findings are described. Such kind of analyses have been published before for other health care settings and larger patient populations. A relatively small sample size is the major limitation and prevents novel insight or meaningful conclusions. Questions regarding use of albumin arise particularly in controversial issues such as combinations of a particular clinical condition with significant hypoalbuminemia with and without hypovolemia or when 5% vs. 20% to 25% human albumin solutions are used. Inappropriate use in non-cardiac surgery would merit further analysis, however, sample size is too small.

---

## [Author Response · Author response to Decision Letter 0]

31 Aug 2021

We believe the results found in this research warrant a publication because there is less data showing how albumin is used overall in a system. I agree with the reviewer that the prescribing practices do not differ between a community hospital setting versus a tertiary hospital setting. While multiple publications have developed criteria for appropriate albumin use, many recent papers note albumin use in specific circumstances. This paper is a look at the distribution of albumin within the current community healthcare setting. It can also serve as a starting point for future research into which departments’ albumin use should be more scrutinized. 

The changes suggested in the first researcher’s review have been applied to the paper as follows: the research works best as a cross-sectional review of how albumin is typically used within a given healthcare system. From these results, other healthcare systems may be able to decide to pursue research into certain departments regarding the necessity of albumin use. The safety data was not reviewed due to the scope of the research project though that would be an interesting follow-up research project. These results mainly show the distribution of albumin use and for what indications is it most appropriate. 

For the second reviewer, we agree that one of the major limitations with this article is the small sample size which prevents novel insight. Our research may be used as a preliminary look at one community hospital setting on which other studies may be conducted to evaluate albumin use in a larger healthcare setting with a wider patient population. 

Thank you again for your time. Please find attached a revised manuscript with the recommended changes.

---

## [Editor Report · Decision Letter 1]

13 Sep 2021

Evaluation of Albumin use in a Community Hospital Setting: A retrospective study looking at appropriate use and prescribing patterns

PONE-D-21-19973R1

Dear Dr. JOHN,

We’re pleased to inform you that your manuscript has been judged scientifically suitable for publication and will be formally accepted for publication once it meets all outstanding technical requirements.

Kind regards,

Aleksandar R. Zivkovic

Academic Editor

PLOS ONE

---

## [Editor Report · Acceptance letter]

28 Sep 2021

PONE-D-21-19973R1 

Evaluation of Albumin use in a Community Hospital Setting: A retrospective study looking at appropriate use and prescribing patterns 

Dear Dr. John:

I'm pleased to inform you that your manuscript has been deemed suitable for publication in PLOS ONE. Congratulations! Your manuscript is now with our production department. 

Kind regards, 

on behalf of

Dr. Aleksandar R. Zivkovic 

Academic Editor

PLOS ONE